

# Characterization of a Portable, Light-Weight, Low-Power Chemical Ionization Time-of-Flight Mass Spectrometer

Austin D. Dobrecevich[1,2], Felipe Hilfiker-Lopez[3], Chris J. Wright[2], Urs Rohner[3], Joel A. Thornton[2]

[1]Department of Chemistry, University of Washington, Seattle, 98195, USA

[2]Department of Atmospheric and Climate Science, University of Washington, Seattle, 98195, USA

[3]Tofwerk AG, Thun, 3645, Switzerland

*Correspondence to*: Joel A. Thornton (joelt@uw.edu)

**Abstract.**

We have developed and characterized the performance of a portable time of flight chemical ionization mass spectrometer

(Portable-TOF-CIMS) capable of detecting trace gases at parts per trillion by volume (pptv) mixing ratios in ambient air. The instrument is compact ($0.063$ m$^3$), weighs less than 30 kg, and operates on 270 W of 24 VDC power. These characteristics allow it to be readily deployed on a range of mobile or stationary platforms with little electrical or structural engineering considerations. The mass spectrometer achieves a mass resolving power of 1300 ($m/\Delta m$) at mass-to-charge of 381 (m*/Q) and* a mass accuracy of <10 *ppm*. The instrument can operate in both positive or negative polarity and therefore can detect a suite

of different analytes depending upon the reagent ion chemistry. We demonstrate the instrument response to inorganic and organic trace gases using Iodide anion adduct and Benzene cation reagent ion chemistries and illustrate its performance sampling ambient air during a multi-week stationary deployment and mobile deployments from two different personal automobiles and from a cargo e-bike using only a battery to power the instrument during operation.

**Short Summary**

A portable, battery-operable mass spectrometer has been developed to probe atmospheric composition. It shows sensitive detection with high time resolution of several atmospheric trace gases relevant to air quality. Operation of the instrument while walking, bicycling, aboard light aircraft, towers, and off-the-grid locations is possible. Deployment across these platforms allows for more routine acquisition of atmospheric chemical composition information across many source types

and locations.



## 1 Introduction

High time and spatial resolution measurements of air pollutants are fundamental to gaining a greater understanding of the sources and impacts of air pollution, such as on human health (Krilaviciute et al., 2015; Sethi et al., 2013; Tsai, 2016) and on atmospheric and ecosystem processes. Chemical Ionization Mass Spectrometry (CIMS) has become a common approach to

measure an array of gas-phase pollutants *in situ*, both indoors and throughout the atmosphere (Berresheim et al., 2000; Bertram et al., 2011; Huey, 2007; Lee et al., 2014). Coupling various chemical ionization schemes to Time-of-Flight mass analyzers (TOF-CIMS), has proven to be well suited for measuring a broad array of ambient organic and inorganic trace gases at high time resolution and signal-to-noise ratio (S/N) with sub parts per trillion (pptv) detection limits (Bertram et al., 2011; Hutterli et al., 2022; Kim et al., 2016; Lee et al., 2014; Murschell et al., 2017). However, the size and power needs of these instruments

make widespread deployment difficult without significant logistical infrastructure including suitable platforms and shelter, power availability and gas supply constraints. This limits the ability to routinely study atmospheric composition, emission sources, and personal exposure at the neighbourhood to regional scales (Klepeis et al., 2009; Seigneur et al., 1983; Wallace, 1991; Yeoman et al., 2020).

A lightweight, compact, and low-power ToF-CIMS could enable broader deployability and improve the temporal and spatial

resolution of atmospheric composition. As this instrumentation platform is more portable, different micro-environments can be explored to address topics like daily human VOC, NOx, and VCP exposure, off-the-grid remote location measurements, mobile vehicle measurements, and light aircraft without significant engineering or power demands. Another benefit of having a small and portable instrument is that it can be positioned very near the sampling source and eliminate the need for longer inlet lines making sticky or reactive compounds measurements less uncertain.

Decreasing the size and power of a ToF-CIMS can have important implications for performance. Most notably, the length of the mass analyser drift region where ion time-of-flight is determined is directly related to the mass resolving power ($m/\Delta m$), and thus the ability to separate isobaric species. The flight tube, while a significant limitation on overall size, contributes marginally to the overall weight and power of the instrument. The weight and power consumption of a typical ToF-CIMS instrument is dominated by the vacuum pumps, electronics, and chamber which are difficult to optimize given the requirement

of a differentially pumped interface including multiple ion guides, high voltage pulsers and power supplies. To realize a compact, lightweight and low power instrument, the gas load of the instrument must be reduced by reducing orifice dimensions in the differentially pumped interface, so that smaller pumps can be used. The reduced target areas of these orifices reduces the acceptance area for ions to transit successive regions of the interface on the way to the mass analyser. To avoid detrimental loss of sensitivity the reduction of diameters is compensated by the efficient use of multiple RF devices which can efficiently

separate ions from the neutral gas which must be removed before ion analysis. The reduced target areas of these pinholes reduces the acceptance area for ions to transit successive regions of the interface on the way to the mass analyser. Therefore, to maintain the benefits of CIMS as a technique (sensitive, fast) a balanced approach must be taken to optimize the weight,



power and performance relative to the analytes of interest. This is of particular importance for atmospheric studies, where the combined effects of dilution of primary emissions and the chemical transformation of directly emitted compounds into reactive

intermediates and particles can result in concentrations from *ppbv* down to and below single digit *pptv* concentrations. Measurements from moving platforms such as aircraft, ships, or vehicles or stationary high frequency flux approaches for which a portable CIMS approach would excel at the very least requires outstanding sensitivity coupled with selective ion chemistry to maintain analytical performance and utility.

Herein, we present an advancement on ToF-CIMS portability and performance. This novel Portable ToF-CIMS weighs < 30

65    kg, requires 270 W of power, and fits within a hand-towable suitcase (Pelican 1610 Protector). The instrument can be operated continuously for ~4 hrs on a typical battery weighing and the runtime can be easily extended during daytime with consumer grade solar panels when remote operation is required for extended periods of time. The instrument is single person portable and achieves a collision-limited sensitivity of, **30** *ncps/pptv*, limits of detection (LOD, 3σ) of 1.2 pptv in 1-minute averages and a mass resolving power of 1300 $m/\Delta m$. We describe the instrument components and characterize its performance in the

laboratory and on a series of deployments in which conventional instruments would be poorly suited to demonstrate the instruments portability, robustness, and ease of use.

## 2 Experimental

### 2.1 Portable-TOF-CIMS Instrument Description

The Portable-TOF-CIMS, shown schematically in Figure 1a and summarized in Table 1, is analogous in overall design to

commercially available ToF-CIMS from Tofwerk AG but is much smaller and lighter, with a vacuum system and electronics package capable of supporting either a Vocus AIM ion molecule reaction region (IMR) (Riva et al., 2024) or a PTR reaction region (Yatsyna et al, in prep, Kretchmer et al.). As such, a summary of the key differences compared to conventional chemical ionization time of flight instruments is presented here.

One critical point of optimization is the instrument vacuum system. The instrument is pumped by dual (MD 1, VACUUBRAND) to hold the reaction pressure of the AIM reactor at 100 mbar, under otherwise standard flow conditions of (standard cubic centimetres per minute) of ambient air mixed with of reagent ion laden flow drawn through the IMR region. The reagent ion is generated and provided by passing a flow of up to 250 standard cubic centimetres per minute (sccm) of Ultra-High Purity (UHP) Nitrogen ($N_2$) through a heated permeation tube filled with a reagent ion precursor solution (in this

work, benzene with trace methyl iodide (<0.5%)), into a region illuminated by a vacuum ultra-violet lamp (VUV). VUV photoionization generates a high intensity reagent ion flow (here, either Benzene cations, $B^+$, or Iodide anions, $I^-$). The flow from the reagent ion source is immediately mixed with the ambient air sample to produce analyte ions by various chemical ionization mechanisms (Riva et al., 2024).



The reacted analyte-reagent ion mixture is subsampled, after ~30 milliseconds of reaction time, through a critical orifice into the first stage of a 3-stage vacuum chamber with all 4 pressure stages evacuated using a single split-flow turbomolecular pump (Pfeiffer SF80). In this first stage, held at by a drag stage of the turbo pump ions are focused by a small RF-only quadrupole ion guide pushed in the axial direction primarily by flow and a weak DC electric field. The focused ion beam passes through another critical orifice into the second stage, held at, where a longer RF-only quadrupole further cools and focuses the ion

beam, after which it passes through a lens stack and finally into the ion extraction region. Ions are pulsed from the extraction region at the Time-of-Flight (TOF) region, where ions fly a V-shaped trajectory to the multi-channel plate (MCP) detectors. The arrival time and intensity are then related to the ion mass-to-charge (Th) and abundance through a digitizer and preamplifier (SP Devices ADQ14-2C). The instrument achieves a mass resolving power of 1300 and mass accuracy of better than 10 ppm.

The instrument autonomously samples and ionizes ambient air continuously and records and displays mass spectra in real-time at a rate. For the analysis and evaluation of the instrument presented here, we used benzene cations ions to detect VOCs and ammonia ($NH_3$) through charge transfer or clustering (Aggarwal et al., 2025; Kim et al., 2016), and $I^-$ adduct ionization for oVOCs, inorganic and organic acids and halogens, among other species. Both reagent ion chemistries generally produce mass

spectra with limited fragmentation and thus lower data complexity which allows many compounds to be quantified accurately even with relatively low mass resolving power. As an example, the detection of gas phase acids, like nitric acid, formic acid, nitrous acid, lactic acid and other mixed organics do not benefit from higher resolution analyzers as the ion chemistry is selective. Similarly, there are few interfering isobars for the detection of typical hydrocarbons like toluene, xylene, trimethylbenzene, isoprene, terpenes, and sesquiterpenes with benzene cations. For these cases, in particular for reactive or

semi-volatile acids the extreme portability eliminates the need for long sample inlets therefor supports more accurate measurements. The instrument operates in either negative or positive ion mode which can be switched through software, taking approximately 1-2 minutes to switch ion detection polarities suitable for monitoring applications. The instrument was calibrated using benzene cations using certified calibration gas mixture comprising of 13 VOCs balanced in nitrogen (Apel-Riemer Environmental, Inc. Miami, FL) with a dynamic dilution using two mass flow controllers (Bronkhorst model: TOF-

101 (30 sccm) and TOF-102 (2000 sccm) full scale). The calibration gas mixture composition is shown in table S1 of the SI.

Recent work has demonstrated that a reagent ion normalized collision-limited sensitivity determined in positive ion mode transfers well to similar chemical ionization schemes in a negative ion mode (Aggarwal et al., 2025). At a fundamental level, the only difference between the two ion mode polarities is the absolute voltage tuning, which could impact absolute ion

transmission between the two polarity ions as all other parameters are constant between the two modes. It follows that then, the collision limited sensitivity normalized to the reagent ion intensities is the same between the two ion modes, assuming that the rate constants are approximately constant between different reagent ions. As hydrocarbons from certified gas standards are



much easier to quantitatively work with which can be accounted for by the reagent ion normalization of calibration factors and detected analytes, we use hydrocarbons to determine the collision limit and therefore the maximum sensitivity of the Portable-
TOF-CIMS.

## 2.2 Portable-TOF-CIMS e-Bike Deployment

As an example of the portability of the Portable-TOF-CIMS we deployed the instrument on an unmodified cargo e-Bike (Figure 1b), with a child carrying basket in the front that could accommodate the instrument as well as a small cylinder of UHP N2
(5.0 UHP Nitrogen, Carbagas, Switzerland). An inlet sample pump drew air at 10 slpm down a ½" PFA sample line 10 cm long. The output of the sample pump was passed through a pressure relief valve before passing through a 20 cm long 4 cm diameter activated carbon trap to generate zero air. This zero air can be delivered via a three-way solenoid valve to programmatically overflow the inlet which provided a consumable-free instrument zero measurement. The instrument was powered with a Jackery Explorer 1000 battery pack (Model Number: JE-1000D, Fremont, CA), which on a single charge can
run the instrument for ~4 hours in sampling mode, and up to 10 hours in full sun with two 200W solar panels. On the handlebars of the bike an anemometer and GPS unit (Maximet GMX500) recorded the wind and location data on the same time resolution as the mass channel signals and was integrated into the TOF data files.

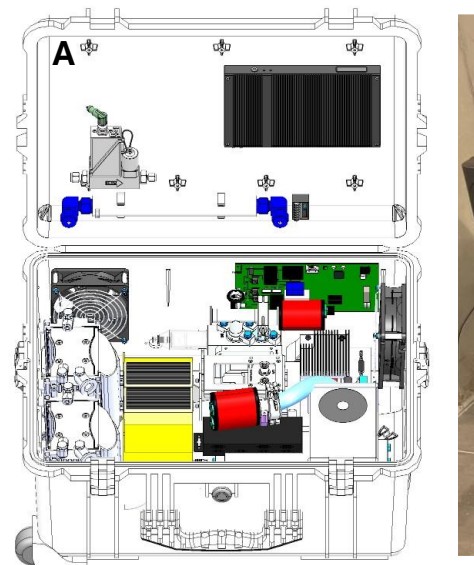
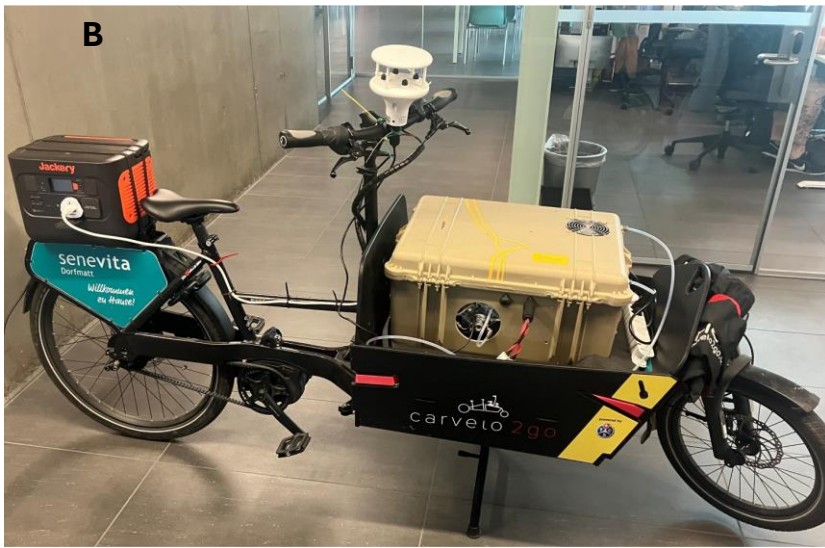

**Figure 1: (a) Model Rendering of the Portable-TOF-CIMS, (b) Portable-TOF-CIMS in standalone operation on a battery pack ready**
**for deployment with cargo e-Bike.**



**Table 1.** Instrument technical and performance characteristics. Normalized ion counts were calculated by dividing all masses at each point by the reagent ion signal, then multiplying by 1 million.

| Characteristic | Information |
|---|---|
| Power | 24 VDC, 270 W |
| Mass | <30 kg |
| Ion-Molecule-Reaction Region | Vocus AIM (Flow Tube CIMS) |
| TOF Size | 25 cm ToF Region |
| Mass Resolution | ~1000-1300 m/$\Delta m$ |
| Normalized Sensitivity (Collision limited) | 30 ncps/ppt |
| Absolute Sensitivity (Collision limited) | 15 cps/ppt |
| $Bz^+$ LOD (α-pinene, 1 min, 3σ) | 1.2 pptv |
| $I^-$ LOD (formic acid, 1 min, 3σ) | 5.0 pptv |

## 2.4 Data Processing

All data was processed using *Tofware* (v. 4.0) in Igor Pro 9 to calibrate the mass axis with a typical mass accuracy of *< 10 ppm* primarily limited by the mass spectral sample frequency (*1 GS/s*). Mass spectral peaks were fitted using mass calibration and Tofware workflows. To account for any variations in reagent ion intensity, the recorded signals at each *m/Q* were divided

by the instantaneous reagent ion signal and then multiplied by a reference value of one million, yielding units of normalized counts per second per million reagent ion counts. The resulting timeseries data, as well as mass spectra were exported from Tofware and plotted with MATLAB R2022b (The Mathworks, Inc., Matick, MA, USA).

## 3 Results and Discussion

Compared to a similarly sensitive commercial ToF-CIMS, the Portable-TOF-CIMS requires a factor of 4X less power,

occupies a factor 16X less space, and weighs a factor of 6 less than the flagship Vocus 2R instrument. These reductions in size, weight, and power, are achieved while maintaining overall analyte sensitivity given that the reagent ion production and chemical ionization conditions remain largely the same. Sufficient ion focusing by the ion optics allows for smaller critical orifices and thus less pumping downstream of the ionization region, such that ion transmission is not affected by the changes in pumping. The primary drawback in the size and power reduction is in mass resolving power (*1300 vs 10,000*).

Instrument sensitivity was determined empirically using a synthetic gas mixture with several analytes of interest, a calibration curve was used to calculate the sensitivity of the instrument to xylene *(m/Q = 106)* in positive ion mode using $Bz^+$ reagent ions





as shown in Fig. 2a-b. For normalization, each mass analyte was divided by the detected reagent ion signal (C6H6+, 78 Th) throughout the time series and then multiplied by one million to normalize the ion signals. Each mixing ratio of xylene was maintained for ~50 sec before changing to a new concentration. Normalized signal was linearly related to the sampled mixing ratio and constrained to pass through zero as shown in Fig. 2b. The resulting slope demonstrates a reagent ion normalized sensitivity to xylene of 30 normalized cps/pptv, which is similar to the expected collision limited sensitivity for the conditions of the Vocus AIM operated at elevated reactor pressure (Aggarwal et al., 2025; Lopez-Hilfiker et al., 2016; Riva et al., 2024). At such high sensitivities the linear range of the instrument is somewhat reduced, with maximum detected analyte concentrations of 50-100 ppbv (if detected at the collision limit).

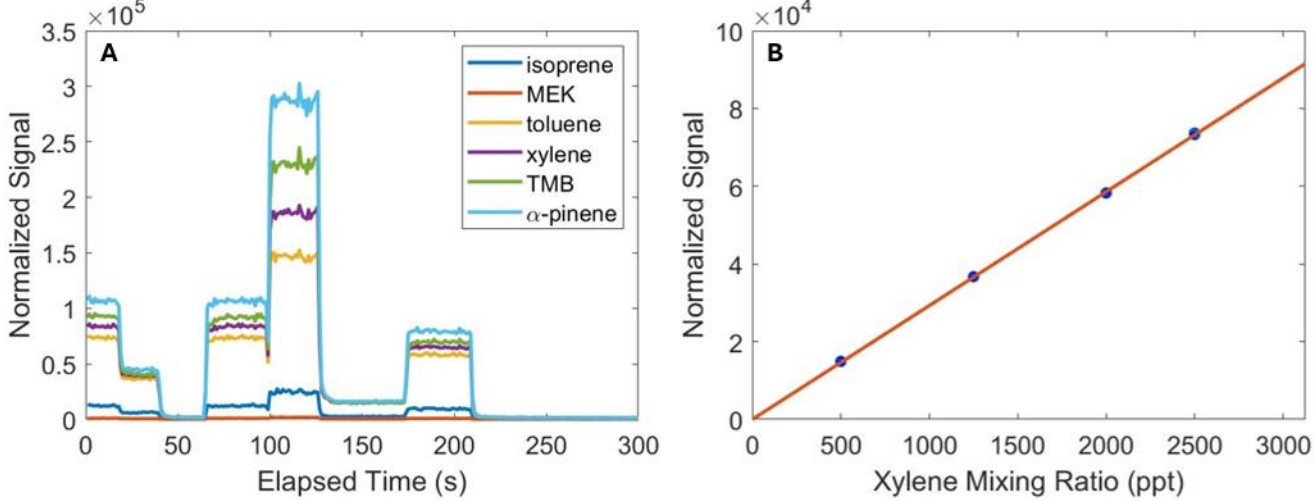

**Figure 2: (a) Calibration sequence using a custom calibration gas mixture in benzene cation ionization mode (normalized by reagent ion signal and multiplied by 1e6), (b) Calibration curve for xylene in benzene cation ionization mode calculated from time averages in 2a.**

In Fig. 3a-e, we illustrate broad features of $Bz^+$ and $I^-$ reagent ion chemistries. Each reagent ion chemistry achieves a relatively low background signal across the target range from 50 to 270 Th (Fig. 3a-d), which ultimately determines detection limits using either reagent ion chemistries. The mass resolving power for the instrument in $I^-$ reagent ion mode ~1300 $m/\Delta m$ shown in Fig. 3e and is representative for positive ion mode as well. A suite of different reagent ion chemistries are possible, such as $NH_4^+$, Acetone cations, protonated Ethanol, $Br^-$, Acetate anions, and Nitrate anions with adjustments to the reagent ion precursors and ion guide voltages, before detection with the time-of-flight mass spectrometer region.




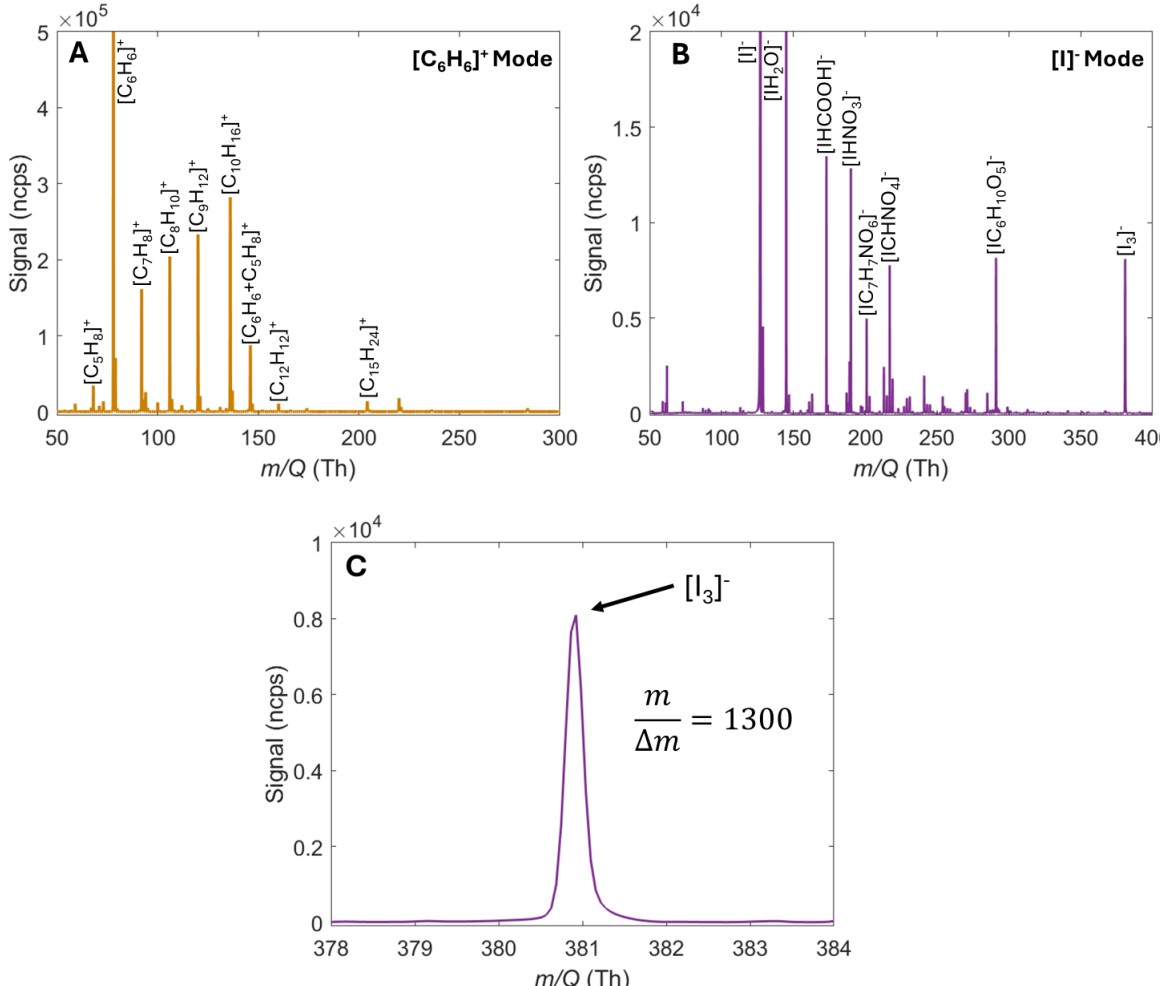

**Figure 3: (a) Normalized high-resolution mass spectrum of a calibration gas over lab air in benzene cation mode with labelled analyte signals. (b) Normalized high-resolution mass spectrum of air measurements in the field in iodide anion mode with labelled analyte signals. (c) Zoom in on I₃⁻ cluster signal illustrating representative peak widths for the high-resolution mass spectrum in Fig. 3b.**

In Fig. 4, we show that with a resolving power of ~1300 Th, and assuming a collision-limited sensitivity of 30 normalized ion counts per pptv, the Portable-TOF-CIMS using $Bz^+$ reagent ion achieves 3σ-LOD in the range of 0.075 to 1.5 pptv for much of the mass spectrum with some individual detection limits higher due to the presence of persistent backgrounds present from reagent ion impurities or reagent ions and water clusters or dimers themselves). In general, at $m/Q$ less than 100 Th the measurement is often background limited (i.e, even the cleanest air has detectable contaminants) while at higher masses (>200 Th) the detection limits become counting statistics dominated (chemical background is negligible).




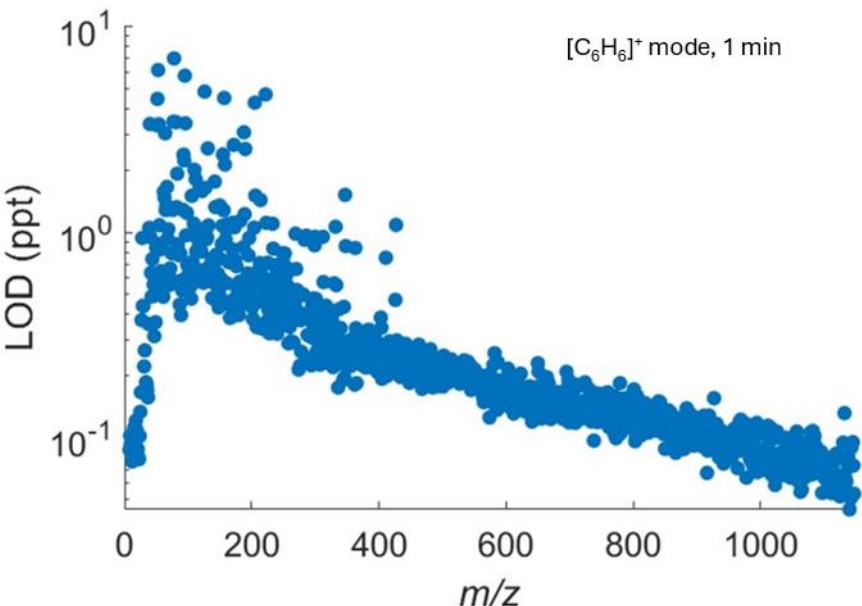

**Figure 4: Limits of detection for chemical ionization using $[C_6H_6]^+$ cations across the mass spectrum assuming ionization is collision**
**limited and a standard sensitivity in one minute of 30 normalized cps/ppt.**

The instrument was operated continuously for 10 days in December using $I^-$ reagent ions and sampling ambient air from the fourth floor of an industrial building in Thun, Switzerland, a small town located in a valley near a lake. Mass spectra were collected in five second averages. In Fig. 5a, we show the normalized ion signal timeseries for the $m/Q$ consistent with the Iodide reagent ion adducts: $IHONO^-$, $IHNO_3^-$, $IC_2H_4O_2^-$, $IHCOOH^-$, $IC_6H_{10}O_5^-$. The nitrous acid adduct signal ($IHONO^-$) was

determined by subtracting the $^{13}C$ isotope from the formic acid adduct ($IHCOOH^-$), and the analytes containing carbon were verified by assessing the expected isotopic distributions. The dynamic range (<1 pptv to > 1 ppbv), ppt-level precision and low detection limits of the instrument are evident. In Fig. 5b, we show the average diurnal cycle of $IHONO^-$ signal as measured by the Portable-TOF-CIMS, which is consistent with that expected for HONO being emitted by automobile traffic and ground elements with a fast daytime loss by photolysis (Kleffmann, 2007; Sörgel et al., 2011).



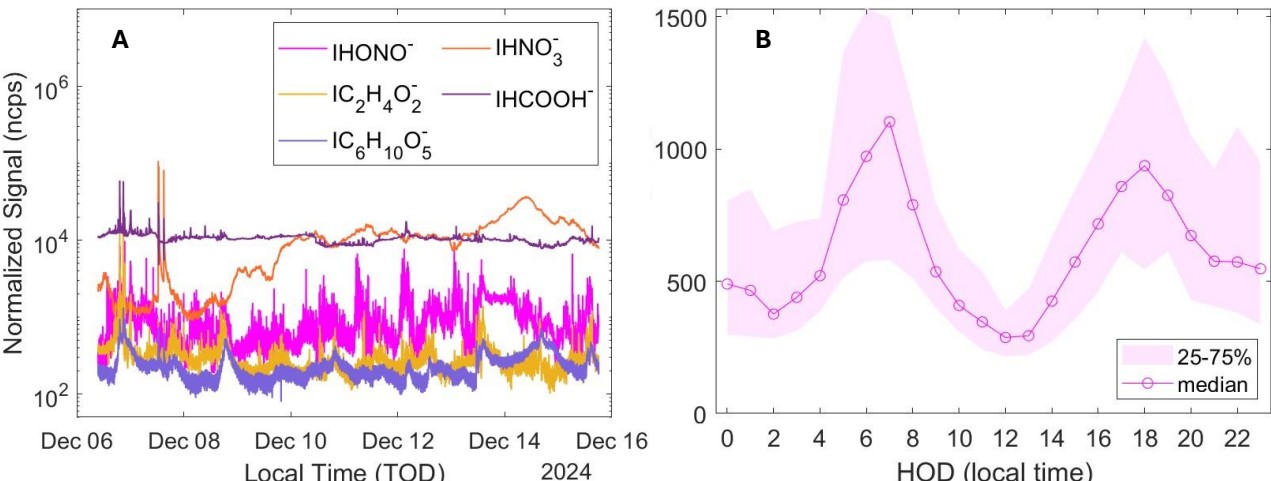

**Figure 5: (a) Stationary measurements of key air pollutants for tracing anthropogenic activity with continuous measurements for 10 days using iodide ion chemical ionization. (b) Diurnal cycle using HONO⁻ Signal averaged hourly.**

The Portable-TOF-CIMS was also deployed in two sampling campaigns using personal automobiles. One campaign was a two-hour drive from Thun to Bern, Switzerland, using the auxiliary battery in a hybrid electric vehicle (Mitsubishi Outlander PHEV) to power the instrument during the drive. The instrument sampled ambient air continuously through an inlet out the passenger window and used I⁻-adduct ionization. Measurements are illustrated in supplementary Fig. S1c, showing dynamic mixing ratio changes of various components along congested city streets indicative of fixed and/or mobile point sources of pollution with high temporal and spatial resolution. The instrument was also deployed in Seattle, WA USA, using a personal passenger vehicle (Subaru Outback PZEV) in Bz+ mode and a standalone battery pack to power the instrument. As shown in supplementary Fig. S2a-b, significant variations in a woodsmoke tracer [$C_7H_8O$, consistent with [Cresol]⁺] detected as a charge transfer product, as well as ammonia, detected as the adduct with B⁺, were measured throughout the drive. These results show the different spatial patterns of these two pollution sources: in residential and lower elevation regions the woodsmoke tracer is elevated, while in more commercial and industrial areas, ammonia is relatively elevated.

To further explore the limits of portability and robustness of the instrument, it was placed on an electric cargo bike, as pictured in Fig. 1b, and powered entirely by a battery pack. The bike was then driven around the city of Thun, Switzerland to probe atmospheric composition and pollution sources along a major bike/walk path following the Aare River, as well as along some major thoroughfares for bus, industrial, and personal vehicles including the central bus station located at Thun, Bahnhof. Normalized signals at several $m/Q$ are plotted in Fig. 6a-d consistent with the I⁻ adducts with HONO, $HNO_3$, $C_6H_{10}O_5$, and $C_{10}H_{15}O_3$, respectively. The latter two $m/Q$ are consistent with that of levoglucosan ($C_6H_{10}O_5$), a known wood smoke tracer commonly detected by I⁻ adduct ionization, and a likely oxidation product of naphthalene $C_{10}H_{15}O_3$), such as an oxo-naphthoquinone, which are known to be emitted by diesel exhaust. In areas with heavy traffic, including around the main bus station and along the main roads, HONO, $HNO_3$, and $C_{10}H_{15}O_3$ related m/Q exhibited elevated signals, consistent with



expectations that these species are derived primarily from anthropogenic pollution. Notably, while both HONO and $HNO_3$ are related to nitrogen oxide emissions, their spatial distributions were not particularly well correlated. This behavior is also consistent with expectations as the formation of $HNO_3$ requires significant atmospheric processing and it has a relatively long lifetime (hours to days), while HONO is mostly directly emitted with a short lifetime (min). Around the train station, indicated by purple arrow in Fig. 6a, HONO mixing ratios were at their highest, indicating an elevated level of fuel combustion emissions in this area (Thun traffic bottleneck), whereas $HNO_3$ reached its highest mixing ratios downwind of the urban areas over larger spatial domains. In contrast, in the waterfront city park with only pedestrian paths, mixing ratios of anthropogenic/vehicle pollution tracers (Fig. 6a-c) were relatively low compared to city streets, except for $C_6H_{10}O_5$ (e.g. levoglucosan). Riding through a lakeside city park, a small campfire was active for grilling, indicated by the orange arrow on Figure 6d and the significantly elevated signal at the m/Q consistent with $C_6H_{10}O_5$ (e.g. levoglucosan). The instrument operated uninterrupted during the ~4-hour e-Bike deployment demonstrating the stability of the instrument electronics, computer control and sampling system even in a relatively primitive mobile platform.

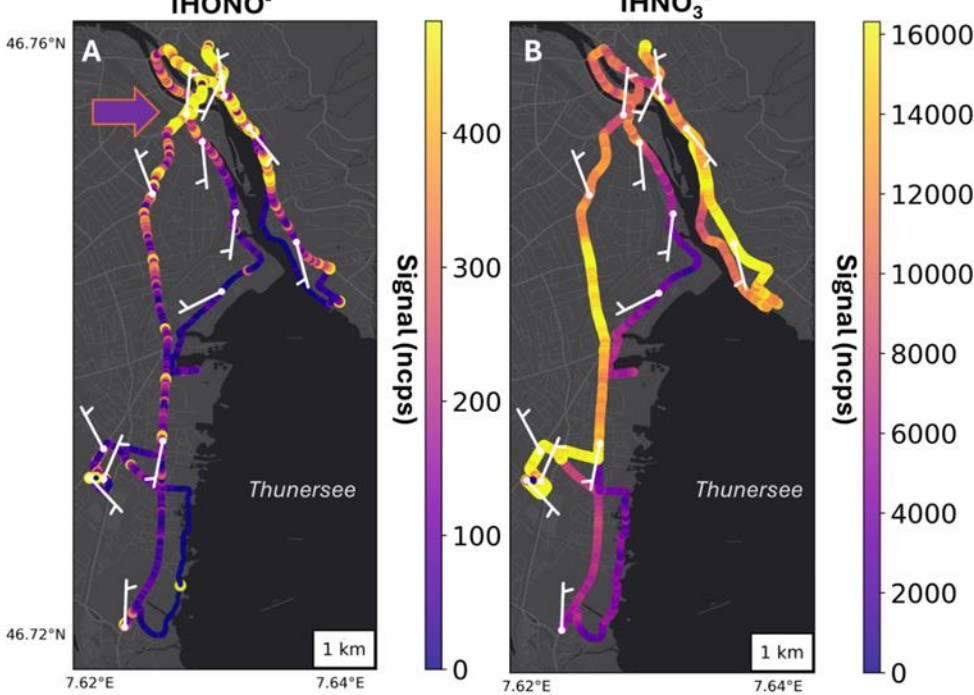



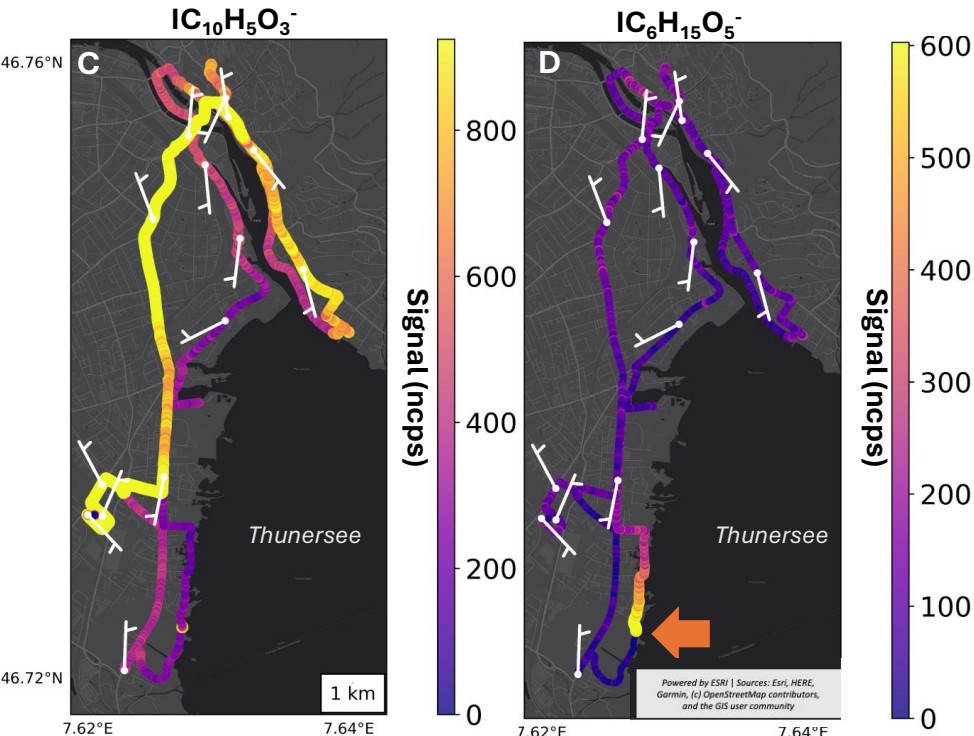

**Figure 6: e-Bike Deployment Data of Portable-TOF-CIMS in Iodide ion Mode overlayed with wind barbs and signal profiles of different key pollution species with colormap in units of normalized cps (a) Signal profile of m/Q consistent with iodide ion adduct with nitrous acid and purple arrow pointing to major bus station in Thun. (b) Signal profile of m/Q consistent with iodide ion adduct with nitric acid. (c) Signal profile of m/Q consistent with $C_{10}H_{15}O_3$ adduct with iodide ion. (d) . Signal profile of m/Q consistent with $C_6H_{10}O_5$ adduct with iodide ion.**

## 4 Conclusion

The Portable-TOF-CIMS shows promise for enhanced portability for air pollution characterization at the point-source to neighbourhood scale, and for enabling deployment in remote environments lacking access to suitable power or infrastructure. The low power needs and compact size allow for operation of the instrument on battery power, as well as in situations where larger size or weight are not possible such as aboard light aircraft, high instrument towers, and remote off-the-grid locations. The instrument was operated in several different environments and showed high sensitivity and stability on par with larger commercially available CIMS instrumentation. *In-situ* measurements made with the instrument are consistent with expected ambient concentrations and reflect expected chemical phenomena for key analytes. The instrument's relatively low mass resolving power ultimately restricts its ability to unambiguously assign molecular composition to detected ions, mandating the



use of selective reagent ion chemistries with minimal fragmentation, or coupling to a chromatography or ion mobility system,
which could negatively affect ease of deployment. Increased efforts for the further miniaturization of the Portable-TOF-CIMS
will be advantageous more widespread and routine measurements of atmospheric or indoor air composition.

## Author Contribution

**Austin D. Dobrecevich:** Methodology, Data curation, Formal analysis, Investigation, Software, Validation, Visualization, Writing - original draft, Writing - review & editing.
**Felipe Hilfiker-Lopez:** Conceptualization, Methodology, Data curation, Formal analysis, Investigation, Validation, Visualization, Writing- review & editing.
**Chris J. Wright:** Data Validation, Visualization, Writing- review & editing.
**Urs Rohner:** Conceptualization, Methodology, Data curation, Formal analysis, Investigation, Validation, Instrument hardware development, Writing- review & editing,
**Joel A. Thornton:** Conceptualization, Investigation, Methodology, Funding acquisition, Project administration, Resources, Supervision, Writing - review & editing.

## Competing Interests

All authors declare that they have no conflict of interest. Felipe Lopez-Hilfiker and Urs Rohner are employees of Tofwerk AG who is a manufacturer & supplier of chemical ionization time of flight mass spectrometers.

## Acknowledgements

The authors thank staff at the University of Washington, in particular Dennis Cannuelle, and Tofwerk AG for their dedication and support of this research.

## Funding Sources

This project was funded by the Arnold and Mabel Beckman Foundation.



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
