# Peer review of "Characterization of a Portable, Light-Weight, Low-Power Chemical Ionization Time-of-Flight Mass Spectrometer"

_EGUsphere, 2025_

## Referee Comment (RC1)

**General Comment**

The manuscript 'Characterisation of a Portable, Light-Weight, Low-Power Chemical Ionization Time-of-Flight Mass Spectrometer' by Dobrecevich et al. introduces a novel, truly portable TOF-CIMS for atmospheric measurements of trace volatiles that can be operated at highest sensitivities. The only presented downside to much bulkier laboratory based or 'transportable' instruments is the reduced mass resolution of only 1300. Nevertheless, with sufficient analytical caution reasonable results can be generated. The authors try to show this with a couple of real world applications.

Although I am deeply convinced by the benefits of a small, light-weight and portable analyzer, I am not entirely satisfied with the analytical characterization of this instrument. My open questions will be addressed in detail in the Specific Comments section. I highly recommend considering these issues with all necessary care to improve the quality of the manuscript.

In addition, there are a plethora of unnecessary mistakes and inconsistencies throughout the manuscript. Follow common guidelines for naming of specific parameters, dimensions, etc. (e.g. IUPAC guidelines of recommendations of nomenclature). Keep the manuscript consistent and do not mix them (e.g. m/z or m/Q or m/Q (Th)). Many of those issues will be highlighted in the Technical Corrections section. Due to time restrictions I can not give a claim of completeness. I recommend careful proofreading prior resubmission.

**Specific Comments**

In the following the Specific Comments section is grouped to major and minor comments. Please note, the major comments are not arranged in order of importance! Minor comments are arranged in order of appearance for easier allocation.

**Major Comments**

**Ionization**

The presented Portable-TOF-CIMS utilizes a Vocus AIM ion-molecule reactor. Being a rather novel technique that certainly differs to previously described TOF-CIMS techniques, there is a limited number of publications available that deal with a thorough characterization of this reduced pressure chemical ionization technique (e.g. Riva et al., 2024, Aggrawal et al., 2025, Song et al., 2026).

Aggrawal et al. (2025) constrain the sensitivity towards an analyte of an AIM IMR as a function of the product ion formation rate and the m/z discrimination, often referred to transmission efficiency (reactor, transfer system and TOF analyzer). The former is defined by the reaction conditions (pressure, temperature, reaction time) as well as reaction rate constants.

However, this does not cover the full picture for a flow-driven CIMS operated at ~100 mbar and reaction times in the range of 10 ms. Here, the reality is much more complex and includes equilibrium reactions, side reactions with alternative reagent ions or clusters of reagent ions. Even potential secondary or multiple generation reactions of initially ionized analytes with analytes of higher ionization energy or cluster affinity are reported. These effects together directly affect sensitivities and induce significant matrix effects.

The proposed ionization pathway is only applicable for another common CIMS technique: A well operated PTR-MS (2-3 mbar, 100 μs reaction time, an E/N to decluster reagent ions) with a significant abundance of reagent ions and only one single ion-molecule reaction that can lead to the ionization of a product (Lindinger et al., 1997). It is well documented that if only one of these parameters is not in the range stated above, the ionization process gets highly nonlinear, especially for complex mixtures like ambient air.

All this is directly visible in Figure 2!

Just a simple question: At ~ 180 - 200 s (Figure 2) the response to the concentration step of all VOCs is virtually the same with ~30k ncps. Now let's compare xylene and TMB, which both are having very similar m/z, polarities and dipole moments and, hence, very similar collisional rates in a buffer gas. When you claim that sensitivity is collisionally limited, how is it possible that the sensitivity changes from about 1:1 at ~ 180 - 200 s to 3:2 at ~100 - 130 s for the calibration point with ~300k ncps?

Also, if xylene is reacting at collisional rate, why is a-pinene showing much higher signals at higher concentrations?

To further illustrate my concerns, I roughly digitized all signals in Figure 2/A and normalized it to the xylene signal intensity:

[Figure]

This figure clearly shows that the whole ionization process is highly nonlinear and actually far away from being collisionally limited.

These effects actually do not come to a surprise and are well known and studied for CIMS operated at pressures and reaction times higher than that of a PTR-MS and, hence, I leave it to the authors digging through the abundant relevant literature on this topic.

As this manuscript is titled 'Characterisation of a Portable, Light-Weight, Low-Power Chemical Ionization Time-of-Flight Mass Spectrometer' and is introducing a novel atmospheric trace gas monitor, it's clearly the scope of this manuscript to address the ionization process properly.

To save the manuscript, consider the following suggestions:

- refrain from the use of (near) collision limited or standard sensitivity. Yes, a single compound could potentially be ionized at collisional limit (at least in B+ mode, I- is reportedly a three body ionization process) and, yes, it could make sense comparing those to assess instrumental parameters (as shown by Aggrawal et al., 2025).
- Add all 6 VOCs to Figure 2/b, cover the entire range (the calibration was done up to 300k ncps whereas only data up to 100k ncps is shown).
- Add the same figure in cps to the supplement and show how the reagent ion is decaying. This can give valuable feedback on linearity when the mixture is not that uniform like in the calibration gas standard.
- Add a discussion on the implications for ambient measurements (matrix effects, linearity, etc.).

**Transmission/Mass Discrimination**

The manuscript introduces the use of smaller orifices, other flows and two RF only ion guides and states '*... ion transmission is not affected by the changes in pumping.*' (**L158**). However, with all changes no change in ion transmission would be highly surprising. This is something that needs to be experimentally proven especially when stated directly like this. This is moreover the case, when a constant transmission with the (above discussed) 'collision limited standard sensitivity' is used to calculate quantitative limits of detections covering a m/z range up to *m/z* 1100 (Figure 4).

A solution could be to only compare the xylene sensitivity for both instruments and show the 3-sigma LODs in units of cps or ncps (for uncalibrated compounds). Make a second plot (or

add a second y-axis) where you add the LODs for the quantified compounds of your calibration mix.

**Minor Comments**

**L81:** Herein the VOCUS AIM is operated at 100 mbar. Riva et al. (2024) introduces standard operation conditions at 50 mbar. Consider discussing the effects of this elevated pressure.

**L105:** According to this statement, the accurate quantification directly follows from a limited amount of fragmentation. Of course, this is not true (see Major Comments), but conservation of one chemical composition after ionization undoubtedly simplifies quantification. However, clusters further complicate qualitative and quantitative analyses (similarly to fragments). Clustered products are mentioned and shown throughout the manuscript.

**L112:** Polarity switching within 1-2 minutes is actually something new. To my knowledge other bipolar VOCUS AIM instruments have 10 min switching times. I recommend removing this from the experimental section and adding it to the results section together with a figure in the supplement that demonstrates this impressively quick polarity switching.

**L132:** How often was this zero delivered? What periods of time are recommended in between those zeros?

**L148:** Why is primarily the sampling rate of 1 ns limiting the mass accuracy? Based on figure 3/C, the peaks seem to be well characterized and additional data points should not improve the peak-centers dramatically.

**L191:** '*...the detection limit becomes counting statistics dominated (chemical background is negligible).*' I do not understand this. The LOD should always be counting statistics dominated as the chemical background is part of it. Please clarify.

**Figure S1.:** Unlike the rest of the examples, quantified data is reported herein. State how the data is quantified and/or show it in ncps.

**Technical Corrections**

**L13:** '*mass-to-charge of 381 (m/Q)*' - follow IUPAC guidelines (https://goldbook.iupac.org/terms/view/M03752): mass-to-charge of *m/z 381.*

**L45:** '*Decreasing the size and power of a TOF-CIMS…*': I assume the power consumption/demand is decreased.

**L46:** '*...the mass analyzer drift region…*': the mass analyzer's drift region

**L52:** '*The reduced target areas of these orifices reduces…*': The reduced target areas of these orifices reduce…

**L55:** repetition of the whole sentence "The reduced target areas... on the way to the mass analyser."

**L58:** change *power* to power consumption

**L66:** chang *hrs* to h following the IUPAC recommendations (https://goldbook.iupac.org/terms/view/H02866)

**L69:** '*...mass resolving power…*': mass resolution. I assume a mass resolution at FWHM is used herein. Please clarify.

**L76:** introduce AIM and PTR

**L77:** The year of the publication for 'Krechmer et al.' is missing (likely 2018). To introduce PTR-MS, cite the original work (e.g. Lindinger et al., 1997). No need to mention work in preparation for such an extensively studied instrument.

**L80 - L85:** There seem to be some inaccuracies. E.g. 'The instrument is pumped by dual…' Dual what? Introduce sccm together with 'standard cubic centimeters per minute' and not in L83. What are the '*otherwise standard flow conditions of (...) of ambient air…*'?

**L86:** B+ is not used consistently throughout the manuscript. Often Bz+ is used. Select the more commonly used nomenclature.

**L94:** '*…, held at, …*': held at what?

**L98:** '*The instrument achieves a mass resolving power of 1300  and mass accuracy of better than 10 ppm.*': This is a result that is also discussed in the result section. Remove it from the experimental section. Also: … and a mass accuracy…

**L102:** '*... at a rate.*': At a rate of what?

**L104:** introduce oVOCs

**L110:** a comma is missing after acids

**L130:** '...*200W...*': 200 W

**L145:** Table 1. Change Power to Power Consumption, stay consistent with units and use pptv instead of ppt. Also note the proper use of the term '*Mass Resolution*' in the table. However, m/delta m is not the unit of the Mass Resolution; it's the definition (m/delta m = 1300; like it is stated in Figure 3/c).

**L147:** The typical mass accuracy is a result. However, please clarify which m/z and which calibration function are used for mass axis calibration?

**L152:** '*MATLAB R2022b (The Mathworks, Inc., Matick, MA, USA)*': here the company is named together with its exact location. Compare to e.g. VACUUBRAND (L81) or Tofwerk AG (L75), where only the company is named. Please unify.

**L154:** for better readability, directly state the flagship VOCUS 2R as the commercial ToF-CIMS. Also: TOF-CIMS or ToF-CIMS? Introduced is TOF-CIMS (L32).

**L172:** Figure 2. add units to y-axis, xylene mixing ratio (ppt) -> (pptv)... needs to be consistent

**L183:** Figure 3. m/Q (Th) -> m/z, [C6H6]+ Mode to B+ or Bz+ (or any other consistent nomenclature); similarly for [I]- Mode.

**L189:** remove ')' at the end of the sentence; change '*...at m/Q less than 100 th*' to *m/z* < 100 (m/z is unitless according to IUPAC guidelines).

**L207:** the caption states HONO- instead of IHONO-

**L226:** '... of naphthalene C10H15O3)': add bracket. Also think of consistency. Is C10H15O3 in agreement with the otherwise used IC10H15O3- or the [IC6H10O5]- format used elsewhere in this manuscript.

**L248:** Figure 6/C titles IC10H5O3-, the caption mentions the C10H15O3 adduct with iodide ion. Figure 6/D titles IC6H15O5- but the caption mentions C6H10O5.

**Table S1:** add the unit of the concentration (I assume ppbv). What is the expected standard error of the stated concentrations? To my knowledge, all calibration gas suppliers state much more accurate concentrations than rounded to the nearest 100 ppbv).

**Figure S1/b:** add a label to the y-axis, not only a unit

**Figure S2.:** add units to the colorbar; be consistent with the labeling (is it NH3+ or is it NH4C6H6+?).

**References**

Lindinger et al. (1997): https://doi.org/10.1016/S0168-1176(97)00281-4

Metzger et al. (2008): https://doi.org/10.5194/acp-8-6453-2008

Kramer et al. (2020): https://doi.org/10.5194/acp-20-5231-2020

Riva et al. (2024): https://doi.org/10.5194/amt-17-5887-2024

Aggrawal et al. (2025): https://doi.org/10.5194/amt-18-4227-2025

Song et al. (2026): https://doi.org/10.1007/s44408-025-00085-z